# Learning Globally Smooth Functions on Manifolds

## Abstract

Smoothness and low dimensional structures play central roles in improving generalization and stability in learning and statistics. The combination of these properties has led to many advances in semi-supervised learning, generative modeling, and control of dynamical systems. However, learning smooth functions is generally challenging, except in simple cases such as learning linear or kernel models. Typical methods are either too conservative, relying on crude upper bounds such as spectral normalization, too lax, penalizing smoothness on average, or too computationally intensive, requiring the solution of large-scale semi-definite programs. These issues are only exacerbated when trying to simultaneously exploit low dimensionality using, e.g., manifolds. This work proposes to overcome these obstacles by combining techniques from semi-infinite constrained learning and manifold regularization. To do so, it shows that, under typical conditions, the problem of learning a Lipschitz continuous function on a manifold is equivalent to a dynamically weighted manifold regularization problem. This observation leads to a practical algorithm based on a weighted Laplacian penalty whose weights are adapted using stochastic gradient techniques. We prove that, under mild conditions, this method estimates the Lipschitz constant of the solution, learning a globally smooth solution as a byproduct. Numerical examples illustrate the advantages of using this method to impose global smoothness on manifolds as opposed to imposing smoothness on average.

## 1 Introduction

Learning smooth functions has been shown to be advantageous in general and is of particular interest in physical systems. This is because of the general observation that close input features tend to be associated with close outputs and of the particular fact that in physical systems Lipschitz continuity of input-output maps translates to stability and safety (Oberman and Calder, 2018; Finlay et al., 2018a; Couellan, 2021; Finlay et al., 2018b; Pauli et al., 2021; Krishnan et al., 2020; Shi et al., 2019; Lindemann et al., 2021; Arghal et al., 2021).

To learn smooth functions one can require the parameterization to be smooth. Such is the idea, e.g., of spectral normalization of weights in neural networks (Miyato et al., 2018; Zhao and Liu, 2020). Smooth parameterizations have the advantage of being globally smooth, but they may be restrictive because they impose smoothness for inputs that are not necessarily realized in the data. This drawback motivates the use of Lipschitz *penalties* in risk minimization (Oberman and Calder, 2018; Finlay et al., 2018a; Couellan, 2021; Pauli et al., 2021; Bungert et al., 2021), which offers the opposite tradeoff. Since penalties encourage but do not enforce small Lipschitz constants, we may learn functions that are smooth on average, but with no global guarantees of smoothness at every point in the support of the data. Formulations that guarantee *global* smoothness can be obtained if the risk minimization problem is modified by the addition of a Lipschitz constant *constraint* (Krishnan et al., 2020; Shi et al., 2019; Lindemann et al., 2021; Arghal et al., 2021). This yields formulations that guarantee Lipschitz smoothness in all possible inputs without the drawback of enforcing smoothness outside of the input data distribution. Several empirical studies (Krishnan et al., 2020; Shi et al., 2019; Lindemann et al., 2021; Arghal et al., 2021) have demonstrated the advantage of imposing global smoothness constraints only on observed inputs.

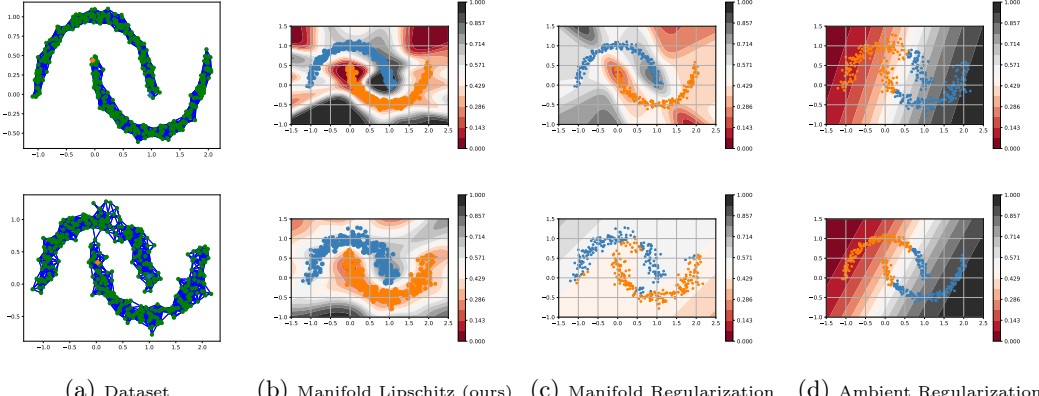

|(a) Dataset|(b) Manifold Lipschitz (ours)|(c) Manifold Regularization|(d) Ambient Regularization|

Figure 1: Two moons dataset. The setting consists of a two dimensional classification problem of two classes with 1 labeled, and 200 unlabeled samples per class. The objective is to correctly classify the 200 unlabeled samples. We consider two cases (top) the estimated manifold has two connected components, and (bottom) the manifold is weakly connected (cf. Figure 1). We plot the output of a one layer neural network trained using Manifold Regularization, Manifold/Ambient Lipschitz. Ambient Regularization fails to classify the unlabeled samples, given that ignores the distribution of samples given by the Manifold. The case in which the manifold has two connected component (cf. Figure 1a), our method works as good as Manifold Regularization, due to the fact that the Lipschitz constant will be made small in both components separately. However, when the manifold is weakly connected, Manifold Regularization fails to recognize the transition between the components, as it will penalize large gradients across the manifold, converging to a plane that connects the two samples. Our Manifold Lipschitz method, as it requires the Lipschitz constant to be small, forces a sharp transition along the point with maximal separation.

In this paper we exploit the fact that data can be often modeled as points in a low-dimensional manifold. We therefore consider manifold Lipschitz constants in which function smoothness is assessed with respect to distances measured over the data manifold (Definition 1). Although this looks like a minor difference, controlling Lipschitz constants over data manifolds is quite different from controlling Lipschitz constants in the ambient space. In Figure 1 we look at a classification problem with classes arranged in two separate half moons. Constraining Lipschitz constants in the ambient space effectively assumes the underlying data is uniformly distributed in space [cf. Figure 1-(d)]. Constraining Lipschitz constants in the data manifold, however, properly accounts for the data distribution [cf. Figure 1-(a)].

This example also illustrates how constraining manifold Lipschitz constants is related to manifold regularization (Belkin et al., 2005; Niyogi, 2013; Li et al., 2022). The difference is that manifold regularization penalizes the average norm of the manifold gradient. This distinction is significant because regularizing is more brittle than imposing constraints. In the example in Figure 1, manifold regularization fails to separate the dataset when the moons are close [cf. Figure 1-(c), bottom]. Classification with a manifold Lipschitz constant constraint is more robust to this change in the data distribution [cf. Figure 1-(a), bottom].

Global constrains in the manifold gradient yield a statistical constrained learning problem with an infinite and dense number of constraints. This is a challenging problem to approximate and solve. Here, we approach the solution of this problem in the Lagrangian dual domain and establish connections with manifold regularization that allow for the use of point cloud Laplacians. Our specific contributions are the following:

(C1) We introduce a constrained statistical risk minimization problem in which we learn a function that: (i) attains a target loss and (ii) attains the smallest possible manifold Lipschitz constant among these functions that satisfy the target loss (Section 2).

(C2) We introduce the Lagrangian dual problem and show that its empirical version is a statistically consistent approximation of the primal. This results does *not* require the learning parametrization to be convex (Section 3.1).

(C3) We generalize results from the manifold regularization literature to show that under regularity conditions, the evaluation of manifold Lipschitz constants can be recast in a more amenable form utilizing a weighted point cloud Laplacian (Proposition 3 in Section 3.2).

(C4) We present a dual ascent algorithm to find optimal multipliers. The function that attains the target loss and minimizes the manifold Lipschitz constant follows as a byproduct (Section 3.3).

(C5) We illustrate the merits of learning with global manifold smoothness guarantees with respect to ambient space and standard manifold regularization through two examples: (i) learning robot navigation policies in a space with obstacles and (ii) learning model mismatches in differential drive steering over non-ideal surfaces (Section 4).

RELATED WORK

This paper is at the intersection of learning with Lipschitz constant constraints (Oberman and Calder, 2018; Finlay et al., 2018a; Couellan, 2021; Pauli et al., 2021; Bungert et al., 2021; Miyato et al., 2018; Zhao and Liu, 2020; Krishnan et al., 2020; Shi et al., 2019; Lindemann et al., 2021; Arghal et al., 2021) and manifold regularization (Belkin et al., 2005; Niyogi, 2013; Li et al., 2022; Hein et al., 2005; Belkin and Niyogi, 2005). Relative to the literature on learning with Lipschitz constraints we offer the ability to leverage data manifolds. Since data manifolds are often characterized with unlabeled data (Kejani et al., 2020; Belkin and Niyogi, 2004; Jiang et al., 2019; Kipf and Welling, 2016; Yang et al., 2016; Zhu, 2005; Lecouat et al., 2018; Ouali et al., 2020; Cabannes et al., 2021), similar to these works, we utilize the point cloud Laplacian technique to compute the integral of the norm of the gradient. Relative to the literature on manifold regularization we offer global smoothness assurances instead of an average penalty of large manifold gradients. In (Krishnan et al., 2020), the space is discretized maintaining convexity, and at the expense of deriving a sensitive solution, their approach leads to a large optimization problem. Similar to us, (Krishnan et al., 2020) poses the problem of minimizing a Lipschtiz constant, however, they utilize a softer surrogate (i.e. $p$-norm loss) which is a smoother version of the Lipschitz constant. Their approach therefore, tradeoffs numerical stability (small $p$) with accurate Lipschitz constant estimation ($p = \infty$). We do not work with surrogates, and we seek to minimize the maximum norm of the gradient, utilizing an epigraph technique.

## 2  GLOBAL CONSTRAINING OF MANIFOLD LIPSCHITZ CONSTANTS

We consider data pairs $(x, y)$ in which the input features $x \in \mathcal{M} \subset \mathbb{R}^D$ lie in a compact oriented Riemannian manifold $\mathcal{M}$ and the output features are real valued $y \in \mathbb{R}$. We study the regression problem of finding a function $f_\theta : \mathcal{M} \to \mathbb{R}$, parameterized by $\theta \in \Theta \subset \mathbb{R}^Q$ that minimizes the expectation of a nonnegative loss $\ell : \mathbb{R} \times \mathbb{R} \to \mathbb{R}_+$, where $\ell(f_\theta(x), y)$ represents the loss of predicting output $f_\theta(x)$ when the world realizes the pair $(x, y)$. Data pairs $(x, y)$ are drawn according to an unknown probability distribution $p(x, y)$ on $\mathcal{M} \times \mathbb{R}$ which we can factor as $p(x, y) = p(x)p(y|x)$.

We are interested in learning smooth functions, i.e. functions with *controlled* variability over the manifold $\mathcal{M}$. We therefore let $\nabla_\mathcal{M} f_\theta(x)$ represent the manifold gradient of $f_\theta$ and introduce the following definition.

**Definition 1** (Manifold Lipschitz Constant). *Given a Riemannian manifold $\mathcal{M}$, the function $f_\theta : \mathcal{M} \to \mathbb{R}$ is said to be L-Lipschitz continuous if there exists a strictly positive constant $L > 0$ such that for all pairs of points $x_1, x_2 \in \mathcal{M}$,*

$$|f_\theta(x_1) - f_\theta(x_2)| \leq L \, d_\mathcal{M}(x_1, x_2), \tag{1}$$

where $d_{\mathcal{M}}(x_1, x_2)$ denotes the distance between $x_1$ and $x_2$ in the manifold $\mathcal{M}$. If the function $f_\theta$ is differentiable, (1) is equivalent to requiring the gradient norm to be bounded by $L$,

$$\|\nabla_{\mathcal{M}} f_\theta(x)\| = \lim_{\delta \to 0} \sup_{\substack{x' \in \mathcal{M} \,:\, x' \neq x, \\ d_{\mathcal{M}}(x,x') \leq \delta}} \frac{|f_\theta(x) - f_\theta(x')|}{d_{\mathcal{M}}(x,x')} \leq L, \quad for\ all\ x \in \mathcal{M}. \qquad (2)$$

With definition 1 in place and restricting attention to differentiable functions $f_\theta$ our stated goal of learning functions $f_\theta$ with controlled variability over the manifold $\mathcal{M}$ can then be written as

$$P^* = \min_{\theta \in \Theta, \rho \geq 0} \rho,$$
$$\text{subject to} \quad \mathbb{E}_{p(x,y)}[\ell(f_\theta(x), y)] \leq \epsilon, \qquad (3)$$
$$\|\nabla_{\mathcal{M}} f_\theta(z)\|^2 \leq \rho, \quad p(z)\text{-a.e.}, \quad z \in \mathcal{M}.$$

In this formulation the statistical loss $\mathbb{E}_{p(x,y)}[\ell(f_\theta(x), y)]$ is required to be below a target level $\epsilon$. Of all functions $f_\theta$ that can satisfy this loss requirement, Problem (3) defines as optimal the one whose Lipschitz constant $L = \sqrt{\rho}$ is the smallest.

The goal of this paper is to develop methodologies to solve (3) when the data distribution and the manifold are unknown. To characterize the distribution we are given sample pairs $(x_i, y_i)$ drawn from the joint distribution $p(x, y)$. To characterize the manifold we are given samples $z_i$ drawn from the marginal distribution $p(x)$. This includes the samples $x_i$ from the (labeled) data pairs $(x_i, y_i)$ and may also include (unlabeled) samples $z_i$.

It is interesting to observe that (3) shows that the problems of manifold regularization (Belkin et al., 2005; Niyogi, 2013; Kejani et al., 2020; Li et al., 2022) and Lipschitz constant control (Oberman and Calder, 2018; Finlay et al., 2018a; Couellan, 2021; Finlay et al., 2018b; Pauli et al., 2021; Krishnan et al., 2020; Shi et al., 2019; Lindemann et al., 2021; Arghal et al., 2021) are related. This connection is important to understand the merit of (3). To explain this better observe that there are three motivations for the problem formulation in (3): (i) It is often the case that if samples $x_1$ and $x_2$ are close, then the conditional distributions $p(y \mid x_1)$ and $p(y \mid x_2)$ are close as well. A function $f_\theta$ with small Lipschitz constant leverages this property. (ii) The Lipschitz constant of $f_\theta$ is *guaranteed* to be smaller than $L = \sqrt{\rho}$. This provides advantages in, e.g., physical systems where Lipschitz constant guarantees translates to stability and safety assurances. (iii) It leverages the intrinsic low-dimensional structure of the manifold $\mathcal{M}$ embedded in the ambient space. In particular, this permits taking advantage of unlabeled data.

Motivations (i) and (ii) are tropes of the Lipschitz regularization literature; e.g., Oberman and Calder (2018); Finlay et al. (2018a); Couellan (2021); Finlay et al. (2018b); Pauli et al. (2021); Krishnan et al. (2020); Shi et al. (2019); Lindemann et al. (2021); Arghal et al. (2021). Indeed, the problem formulation in (3) is inspired in similar problem formulations in which the Lipschitz constant is regularized in the ambient space,

$$\min_{\theta \in \Theta, \rho \geq 0} L,$$
$$\text{subject to} \quad \mathbb{E}_{p(x,y)}[\ell(f_\theta(x), y)] \leq \epsilon, \qquad (4)$$
$$|f_\theta(y) - f_\theta(z)| \leq L\|y - z\|, \quad (y, z) \sim p(x).$$

A difference between (3) and (4) is that in the latter we use a Lipschitz condition that does not require differentiability. A more important difference is that in (4) the Lipschitz constant is regularized in the ambient space. The distance between features $y$ and $z$ in (4) is the Euclidean distance $\|y - z\|$. This is disparate from the manifold metric $d_{\mathcal{M}}(y, z)$ that is implicit in the manifold gradient constraint in (3). Thus, the formulation in (3) improves upon (4) because of it leverages the structure of the manifold $\mathcal{M}$ [cf. Motivation (iii)].

Motivations (i) and (iii) are themes of the manifold regularization literature (Belkin et al., 2005; Niyogi, 2013; Kejani et al., 2020; Li et al., 2022). And, indeed, it is ready to conclude by invoking Green's first identity (see Section 3.2) that the formulation in (3) is also inspired in the manifold regularization problem,

$$\min_{\theta \in \Theta} \mathbb{E}_{p(x,y)}[\ell(f_\theta(x), y)] + \gamma \int_{\mathcal{M}} \|\nabla_{\mathcal{M}} f_\theta(z)\|^2 p(z) dV(z). \qquad (5)$$

The difference between (3) and (5) is that in the latter the manifold Lipschitz constant is added as a regularization penalty. This is disparate from the imposition of a manifold Lipschitz constraint in (3). The regularization in (5) *favors* solutions with small Lipschitz constant by penalizing large Lipschitz constants, while the constraint in (3) *guarantees* that the Lipschitz constant is bounded by $\rho$. This is the distinction between *regularizing* a Lipschitz constant versus *constraining* a Lipschitz constant. The constraint in (5) is also imposed at *all* points in the manifold, whereas the regularization in (5) is an average over the manifold. Taking an average allows for large Lipschitz constants at some specific points if this is canceled out by small Lipschitz constants in other points of the manifold. Both of these observations imply that (3) improves upon (5) because it offers global smoothness guarantees that are important in, e.g., physical systems [cf. Motivation (iii)].

**Remark 1** (Alternative Manifold Lipschitz Control Formulations). There are three arbitrary choices in (3). (a) We choose to constrain the average statistical loss $\mathbb{E}_{p(x,y)}[\ell\,(f_\theta(x),y)] \leq \epsilon$; (b) we choose to constrain the pointwise Lipschitz constant $\|\nabla_{\mathcal{M}} f_\theta(z)\|^2 \leq \rho$; and (c) we choose as our objective to require a target loss $\epsilon$ and minimize the Lipschitz constant $L = \sqrt{\rho}$. We can alternatively choose to constrain the pointwise loss $\ell\,(f_\theta(x),y) \leq \epsilon$, to constrain the average Lipschitz constant $\int_{\mathcal{M}} \|\nabla_{\mathcal{M}} f_\theta(z)\|^2 p(z) dV(z) \leq \rho$ [cf (5)], or to require a target smoothness $L = \sqrt{\rho}$ and minimize the loss $\epsilon$. All of the possible eight combinations of choices are of interest. We formulate (3) because it is the most natural intersection between the regularization of Lipschitz constants in ambient spaces [cf. (4)] and manifold regularization [cf. (5)]. The techniques we develop in this paper can be adapted to any of the other seven alternative formulations.

## 3 LEARNING WITH GLOBAL LIPSCHITZ CONSTRAINTS

Problem (3) is a constrained learning problem that we will solve in the dual domain (Chamon and Ribeiro, 2020). To that end, observe that (3) has statistical and pointwise constraints. The loss constraint $\mathbb{E}_{p(x,y)}[\ell\,(f_\theta(x),y)] \leq \epsilon$ is said to be statistical because it restricts the expected loss over the data distribution. The Lipschitz constant constraints $\|\nabla_{\mathcal{M}} f_\theta(z)\|^2 \leq \rho$ are said to be pointwise because they are imposed for all individual points in the manifold except for a set of zero measure. Consider then a Lagrange multiplier $\mu$ associated with the statistical constraint $\mathbb{E}_{p(x,y)}[\ell\,(f_\theta(x),y)] \leq \epsilon$ and a Lagrange multiplier distribution $\lambda(z)$ associated with the set of pointwise constraints $\|\nabla_{\mathcal{M}} f_\theta(z)\|^2 \leq \rho$. The dual problem associated with (3) can then be written as

$$D^* = \max_{\mu,\lambda \geq 0} \min_\theta L(\theta,\mu,\lambda) := \mu\Big(\mathbb{E}[\ell\big(f_\theta(x),y\big)] - \epsilon\Big) + \int_{\mathcal{M}} \lambda(z)\|\nabla_{\mathcal{M}} f_\theta(z)\|^2 p(z) dV(z),$$

$$\text{subject to} \quad \int_{\mathcal{M}} \lambda(z) p(z) dV(z) = 1. \tag{6}$$

We point out that in (6) we remove $\rho$ from the Lagrangian by incorporating the dual variable constraint $\int_{\mathcal{M}} \lambda(x) p(x) dV(x) = 1$; see Appendix A for details.

We henceforth use the dual problem (6) in lieu of (3). Since we are interested in situations in which we do not have access to the data distribution $p(x,y)$ we further consider empirical versions of (6). Consider then $N$ i.i.d. samples $(x_n,y_n)$ drawn from $p(x,y)$ and define the empirical dual problem as,

$$\hat{D}^\star = \max_{\hat{\mu},\hat{\lambda} \geq 0} \min_\theta \hat{L}(\theta,\hat{\mu},\hat{\lambda}) := \hat{\mu}\bigg(\frac{1}{N}\sum_{n=1}^N \ell(f_\theta(x_n),y_n) - \epsilon\bigg) + \frac{1}{N}\sum_{n=1}^N \hat{\lambda}(x_n)\|\nabla_{\mathcal{M}} f_\theta(x_n)\|^2,$$

$$\text{subject to} \quad \frac{1}{N}\sum_{n=1}^N \hat{\lambda}(x_n) = 1, \tag{7}$$

where, to simplify notation, we assume no unlabeled samples are available. If unlabeled samples are given the modification is straight-forward; see Appendix B.

The remainder of this section provides three technical contributions:

(C2) To justify solutions of (7) we must show statistical consistency with respect to the primal problem (3). This is challenging because of two reasons: (i) Since we did not assume the use of a convex parameterization, (3) is not a convex problem on $\theta$. Thus, the primal and dual problems are not necessarily equivalent. (ii) Since we are maximizing over the dual variables $\mu$ and $\lambda(z)$, we do not know if the empirical dual formulation in (7) is close to the statistical dual formulation in (6). We will overcome these two challenges and show that the empirical dual problem (7) is a consistent approximation of the statistical primal problem (Proposition 1 in Section 3.1).

(C3) Solving (7) requires evaluating the gradient norm sum $(1/N) \sum_{n=1}^{N} \hat{\lambda}(x_n) \| \nabla_{\mathcal{M}} f_\theta(x_n) \|^2$. We will generalize results from the manifold regularization literature to show that under regularity conditions on $\lambda$ the gradient norm integral can be computed in a more amenable form utilizing a weighted point cloud Laplacian (Proposition 3 in Section3.2).

(C4) We introduce a primal-dual algorithm to solve (7) (Section 3.3).

## 3.1 Statistical consistency of the empirical dual problem

We show in this section that (7) is close to (3). Doing so requires the following assumptions:

**Assumption 1.** *The loss $\ell$ is $M$-Lipschitz continuous, $B$-bounded, and convex.*

**Assumption 2.** *Let $\mathcal{H} = \{ f_\theta \mid \theta \in \Theta \subset \mathbb{R}^Q \}$ with compact $\Theta$ be the hypothesis class, and let $\bar{\mathcal{H}} = \overline{\mathrm{conv}}(\mathcal{H})$ be the closure of its convex hull. For each $\nu > 0$ and $\varphi \in \bar{\mathcal{H}}$ there exists $\theta \in \Theta$ such that simultaneously $\sup_{z \in \mathcal{M}} |\varphi(z) - f_\theta(z)| \leq \nu$ and $\sup_{z \in \mathcal{M}} \| \nabla_{\mathcal{M}} \varphi(z) - \nabla_{\mathcal{M}} f_\theta(z) \| \leq \nu$.*

**Assumption 3.** *The hypothesis class $\mathcal{H}$ is $G$-Lipschitz in its gradients, i.e. $\| \nabla_{\mathcal{M}} f_{\theta_1}(z) - \nabla_{\mathcal{M}} f_{\theta_2}(z) \| \leq G | f_{\theta_1}(z) - f_{\theta_2}(z) |$.*

**Assumption 4.** *There exists $\zeta(N, \delta) \geq 0$, and $\hat{\zeta}(N, \delta) \geq 0$ monotonically decreasing with $N$, such that*

$$\left| \mathbb{E}[\ell(f_\theta(x), y)] - \frac{1}{N} \sum_{n=1}^{N} \ell(f_\theta(x_n), y_n) \right| \leq \zeta(N, \delta), \left| \mathbb{E}[f_\theta(x)] - \frac{1}{N} \sum_{n=1}^{N} f_\theta(x_n) \right| \leq \hat{\zeta}(N, \delta), \quad (8)$$

*for all $\theta \in \Theta$, with probability $1 - \delta$ over independent draws $(x_n, y_n) \sim p$.*

**Assumption 5.** *There exists a feasible solution $\tilde{\theta} \in \Theta$ such that $\mathbb{E}[\ell(f_{\tilde{\theta}}(x), y)] < \epsilon - M\nu$.*

Assumption 1 holds for most losses utilized in practice. Assumption 2 is a requirement on the richness of the parametrization. In the particular case of neural networks the covering constant $\nu$ is upper bounded by the universal approximation bound of the neural network. Assumption 3 also holds for neural networks if we use smooth nonlinearities – e.g., hyperbolic tangents. The *uniform convergence* property (8) is customary in learning theory to prove PAC learnability and is implied by bounds on complexity measures such as VC dimension or Rademacher complexity (Mohri et al., 2018; Vapnik, 1999; Shalev-Shwartz and Ben-David, 2014). The following theorem provides the desired bound.

**Proposition 1.** *Let $\hat{\mu}^\star, \hat{\lambda}^\star$ be solutions of the empirical dual problem (7). Under assumptions 1–5, there exists $\hat{\theta}^\star \in \mathrm{argmin}_\theta \hat{L}(\theta, \hat{\mu}^\star, \hat{\lambda}^\star)$ such that, with probability $1 - 5\delta$,*

$$|P^\star - \hat{D}^\star| \leq \mathcal{O}(\nu) + (1 + \Delta)\zeta(N, \delta) + \mathcal{O}(\hat{\zeta}(N, \delta)), \quad (9)$$

*where $\Delta = \max(\hat{\mu}^\star, \mu^\star) \leq C$ for a constant $C < \infty$, where $\mu^\star, \hat{\mu}^\star$ are solutions of (6), and (7) respectively.*

Proposition 1 shows that the empirical dual problem 7 is statistically consistent. That is to say, for any realization of $N$ according to $p$, the difference between the empirical dual problem (7) and the statistical smooth learning problem (3) decreases as $N$ increases. This difference is bounded in terms of the richness of the parametrization ($\nu$), the difficulty of the fit requirement (as expressed by the optimal dual variables $\mu^\star, \hat{\mu}^\star$), and the number of samples ($N$). The guarantee has a form typical for constrained learning problems (Chamon and Ribeiro, 2020). Proposition 1 states that we are able to predict what is the minimum norm of the gradient that a function class can have while satisfying an expected $\epsilon$ loss. This

is important, because we do not require access to the distribution of the samples $p$, only a set of $N$ samples sampled according to this distribution. On the other hand, Proposition 1 does not state that by solving the dual problem 7 we will obtain a solution of the primal problem 3. The following proposition provides a bound on the near feasibility of the solution of 7 with respect to the solution of 3.

**Proposition 2.** *Let $\hat{\mu}^\star, \hat{\lambda}^\star$ be solutions of the empirical dual problem (7). Under assumptions 1–5, there exists $\hat{\theta}^\star \in \arg\min_\theta \hat{L}(\theta, \hat{\mu}^\star, \hat{\lambda}^\star)$ such that, with probability $1 - 5\delta$,*

$$\max_{z \in \mathcal{M}} \|\nabla_{\mathcal{M}} f_{\hat{\theta}^\star}(z)\|^2 \leq P^* + \hat{\mu} \left| \frac{1}{N} \sum_{n=1}^{N} \ell(f_{\hat{\theta}^*}(x_n), y_n) - \epsilon \right| + \mathcal{O}(\nu) + \mathcal{O}(\hat{\zeta}(N, \delta)), \text{ and} \quad (10)$$

$$\mathbb{E}[\ell(f_{\hat{\theta}^\star}(x), y)] \leq \epsilon + \zeta(N, \delta). \quad (11)$$

Proposition 2 provides near optimality, and near feasibility conditions for solutions $\hat{\theta}^*$ obtained through the empirical dual problem (7). The difference between the maximum gradient of the obtained solution $\theta$ and the optimal value $P^*$ is bounded by the number of samples $N$, as well as the empirical constraint violation. Notice that even though the optimal dual variable $\hat{\mu}$ is not known, the constraint violation can be evaluated in practice, as it only requires to evaluate the obtained function over the $N$ given samples.

**Remark 2** (Interpolators). *In practice, the number of parameters in a parametric function (e.g. Neural Network) tends to exceed the dimension of the input, which allows functions to interpolate the data, i.e. to attain zero loss on the dataset. Proposition 2, presents a connection to interpolating functions. By setting $\epsilon = 0$, if the function achieves zero error over the empirical distribution i.e. $\ell(f_{\hat{\theta}^*}(x_n), y_n) = 0$, for all $n \in [N]$, then the dependency on $\mu^*$ disappears. This implies that over interpolating classifiers, the one with the minimum Lipschitz constant over the samples, is probably the one that attains the minimum Lipschitz constant over the whole manifold.*

## 3.2 FROM MANIFOLD GRADIENT TO DISCRETE LAPLACIAN

We derive an alternative way of computing the integral of the norm of the gradient utilizing samples. To do so, we define the normalized point cloud Laplacian according a probability distribution $\lambda$.

**Definition 2** (Point-cloud Laplacian). *Consider a set of points $x_1, \ldots, x_N \in \mathcal{M}$, sampled according to probability $\lambda : \mathcal{M} \to \mathbb{R}$. The normalized graph Laplacian of $f_\theta$ at $z \in \mathcal{M}$ is defined as*

$$\mathbf{L}_{\lambda,N}^t f_\theta(z) = \frac{1}{N} \sum_{n=1}^{N} W(z, x_n)\big(f_\theta(z) - f_\theta(x_n)\big), \quad (12)$$

*for*

$$W(z, x_n) = \frac{1}{t} \frac{G_t(z, x_n)}{\sqrt{\hat{w}(z)\hat{W}(x_n)}}, \quad \text{with} \quad G_t(z, x_n) = \frac{1}{(4\pi t)^{d/2}} e^{-\frac{\|z - x_n\|^2}{4t}}, \quad (13)$$

$$\hat{w}(z) = \frac{1}{N} \sum_{n=1}^{N} G_t(z, x_n), \quad \text{and} \quad \hat{W}(x_n) = \frac{1}{N-1} \sum_{m \neq n} G_t(x_m, x_n). \quad (14)$$

As long as the function considered is smooth enough, the following convergence result holds:

**Proposition 3** (Point Cloud Estimate). *Let $\Lambda$ be the set of probability distributions defined a compact $d$-dimensional differentiable manifold $\mathcal{M}$ isometrically embedded in $\mathbb{R}^D$ such that $\Lambda = \{\lambda : 0 < a \leq \lambda(z) \leq b < \infty, |\frac{\partial \lambda}{\partial x}| \leq c < \infty \text{ and } |\frac{\partial^2 \lambda}{\partial x^2}| \leq d < \infty \text{ for all } z \in \mathcal{M}\}$, and let $f_\theta$, with $\theta \in \Theta$, be a family of functions with uniformly bounded derivatives up to order 3 vanishing at the boundary $\partial \mathcal{M}$. For any $\epsilon > 0, \delta > 0$, for all $N > N_0$,*

$$P\left[ \sup_{\lambda \in \Lambda, \theta \in \Theta} \left| \int \|\nabla_{\mathcal{M}} f_\theta(z)\|^2 \lambda(z) dV(z) - \frac{1}{N} \sum_{i=1}^{N} f_\theta(z_i) \mathbf{L}_{\lambda,N}^t f_\theta(z_i) \lambda(z_i) \right| > \epsilon \right] \leq \delta \quad (15)$$

*where the point cloud laplacian $\mathbf{L}_{\lambda,N}^t f_\theta$ is as defined in 2, with $t = N^{-\frac{1}{d+2+\alpha}}$ for any $\alpha > 0$.*

The proof of Proposition 3 relies on two steps. First, we relate the integral of the norm of the gradient over the manifold with the integral of the continuous Laplace-Beltrami operator by virtue of Green's identity. Second, we approximate the value of the Laplace-Beltrami operator by the point-cloud Laplacian. Proposition 3 connects the integral of the norm of the gradient of function $f_\theta$ with a point cloud Laplacian operator, this result connects the dual problem 7, with the primal 3 while allowing for a more amenable way of computing the integral.

**Remark 3** (Laplacian Regularization and Manifold Lipschitz). *The dual problem (6) is closely related to the manifold regularization problem (5). In particular, the two become equivalent by substituting $\rho = \mu^{-1}$, and utilizing a uniform distribution for $\lambda$. The key difference between the problems is given by the dual variable $\lambda$, which can be though as a probability distribution over the manifold that penalizes regions of the manifold where the norm of the gradient of $f_\theta$ is larger. The standard procedure in Laplacian regularization is to calculate the graph-Laplacian of the set of points $\mathbf{L}$ utilizing the heat kernel, and compute the integral by $\mathbf{f}^T \mathbf{L} \mathbf{f}$, where $\mathbf{f} = [f_\theta(x_1), \ldots, f_\theta(x_n)]^T$. In the case of Manifold Lipschitz, the same product can be computed, but utilizing the re-weighted point cloud laplacian.*

### 3.3 Dual Ascent Algorithm

We outline an iterative and empirical primal-dual algorithm to solve the dual problem. Upon the initialization of $\theta_0$, and $\lambda_0, \mu_0$, we set to minimize the dual function as follows,

$$\theta_{k+1} = \theta_k - \eta_\theta \nabla_\theta L(\theta_k, \mu_k, \lambda_k), \tag{16}$$

where $\eta_\theta$ is a positive step-size. Note that to solve problem (16), we can either utilize the gradient version the the lagrangian (cf. equation (7)), or the point-cloud Laplacian (cf. equation (15)). Consequent to updating $\theta$, we update the dual variables as follows,

$$\mu_{k+1} = \left[ \mu_k + \eta_\mu \left( \frac{1}{N} \sum_{n=1}^{N} \ell(f_\theta(x_n), y_n) - \epsilon \right) \right]_+, \tag{17}$$

$$\tilde{\lambda}_{k+1}(x_n) = \lambda_k(x_n) + \eta_\lambda \|\nabla_\mathcal{M} f_\theta(x_n)\|^2 \text{ for all data points } x_n \tag{18}$$

$$\lambda_{k+1} = \operatorname*{argmin}_\lambda \|\tilde{\lambda}_{k+1} - \lambda_{k+1}\|, \text{ such that } \sum_{n=1}^{N} \lambda_{k+1}(x_n) = N. \tag{19}$$

where $\eta_\mu, \eta_\lambda$ are a positive step-sizes. Note that we require a convex projection over $\tilde{\lambda}$ to satisfy the normalizing constraint. In step (18) we require to estimate the norm of the gradient at data point $x_n$, which we estimate for the neighboring data points. Intuitively, the primal dual procedure increases the value of $\lambda(x_n)$ at points in which the norm of the gradient is larger. The role of $\mu$ is to enforce the loss $\ell$ to be smaller than $\epsilon$, a larger value of $\mu$ would increase the relative importance over the norm of the integral. See Appendix F.

## 4 Numerical Results

### 4.1 Navigation Controls Problem

In this section, we consider the problem of continuous navigation of an agent. The agent's objective is to reach a goal while avoiding obstacles. The state space of the agent is given by its position, and navigates by taking actions on the velocity. We construct a square grid of points in the free space i.e. outside of the obstacles, and find the shortest path between two starting positions and the goal along the grid. For those two grid trajectories, we compute the optimal actions to be taken at each point in order to follow the trajectory. The learner is equipped with both labeled trajectories, as well as the unlabeled point grid. To leverage the manifold

| Method | Trajectories |
|---|---|
| ERM | 85 |
| Ambient Reg. | 66 |
| Manifold Reg. | 77 |
| Manifold Lipschitz | 94 |

Table 1: Number of successful trajectories from 100 random starting points.

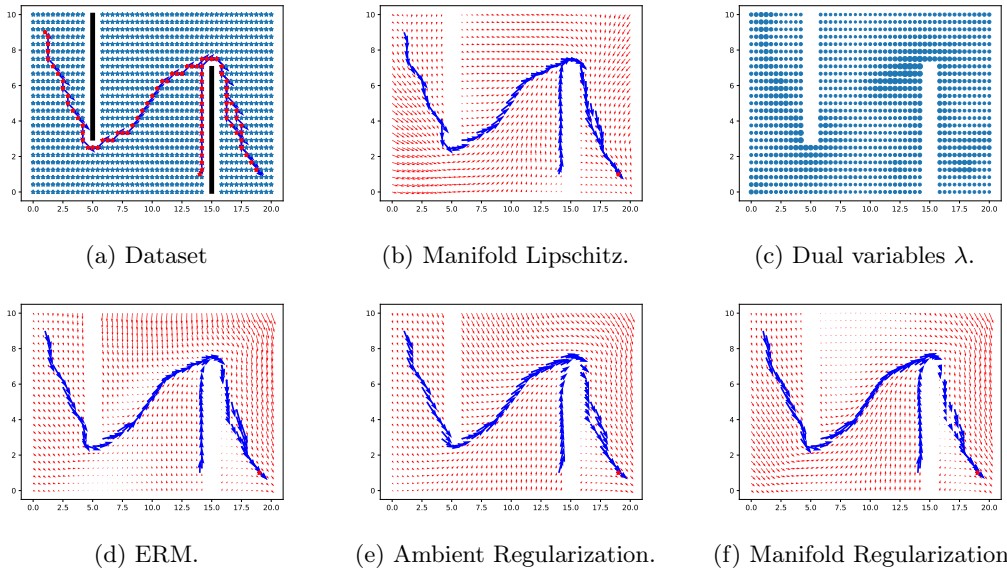

|  | (a) Dataset | (b) Manifold Lipschitz. | (c) Dual variables $\lambda$. |
|---|---|---|---|
|  | (d) ERM. | (e) Ambient Regularization. | (f) Manifold Regularization. |

Figure 2: Figure 2a shows the training dataset, blue stars depict unlabeled point, and blue arrow the optimal action at the red star. Figure 2b shows the learned function using Manifold Learning, and 2c its associated dual variables associated. Figures 2d, 2e, 2f show the functions learned using ERM, ambient regularization, and Manifold regularization respectively.

structure of the data, we consider the grid of points,
and we construct the point cloud Laplacian considering adjacent points in the grid. As measure of merit, we take 100 random points and we compute the trajectories. A trajectory is successful if it achieves the goal without colliding. The results are shown in Table 1, and the learned functions in Figure 2.

## 4.2 Ground Robot Error Prediction

In this real world experiment, we seek to learn the error that a model would make in predicting the dynamics of a ground robot Koppel et al. (2016). We posses trajectories, in the form of time series of the control signals of an *iRobot* Packbot making turns on both pavement, and grass. For each trajectory, the mismatch between the real and the model-predicted states is quantified. Given a dataset of trajectories and errors made by the model, the objective is to learn the error that a model will make. Details of the experiment can be found in G.3.

| Method | Grass | Pavement |
|---|---|---|
| ERM | 0.42 | 0.0120 |
| Ambient Reg. | 0.31 | 0.0065 |
| Manifold Reg. | 0.38 | 0.0045 |
| Manifold Lipschitz | 0.25 | 0.0032 |

Table 2: Error Prediction accuracy for the Ground Robot Experiment.

## 5 Conclusion

In this work, we presented a constrain learning method to obtain smooth functions over manifold data. We have shown that under mild conditions, the problem of finding smooth functions over a manifold can be reformulated as a weighted point cloud Laplacian penalty over varying probability distributions whose dynamics are govern by the constraint violations. Three numerical examples validate the empirical advantages of obtaining functions that vary smoothly over the data.

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
