# OpenReview forum: "Learning Globally Smooth Functions on Manifolds"
_ICLR.cc/2023/Conference — Submitted to ICLR 2023_

### Official Review · Reviewer_2RoL · 2022-10-23

**Confidence:** 3
**Correctness:** 4
**Technical Novelty And Significance:** 3
**Empirical Novelty And Significance:** 2
**Recommendation:** 6

**Clarity, Quality, Novelty And Reproducibility:**

The proposed idea seems to be an extension of the method from the Euclidean space on manifolds. This constitutes a potentially significant contribution, bit I think that the empirical evaluation should be extended. The paper is well written and there are only few spots where it is unclear.

**Strength And Weaknesses:**

There are many parts of the paper that I like e.g. explaining some steps in good detail, but there are few parts which are a bit unclear to me. The introduction of the problem is ok, relying on the example in Fig. 1 and on the difference between favour vs guarantee that a function is globally Lipschitz. Maybe you could elaborate a bit more why using the manifold information is better. The technical content seems reasonable, but I have not checked all the derivations in details. I think that additional experiments should be conducted.

**Summary Of The Paper:**

The authors propose a technique to learn a globally Lipschitz function when the data lie near a manifold. They derive a suitable constrained optimization problem together with the associated optimization scheme. Theoretical results are provided which show that under some conditions it is sensible to solve the empirical optimization problem as it approximates the true problem. The effectiveness of the approach is demonstrated in the experiments.

**Summary Of The Review:**

Questions:
1. In Euclidean spaces it is known that if the norm of the gradient is bounded then the function is Lipschitz. Does the same result hold when $f$ is defined on Riemannian manifolds?

2. I think that the efficiency of your method highly depends on the construction of the graph Laplacian, and for this reason some discussion and demonstrations should be included.

3. Similarly, I think that the experimental setting is a little bit short compared to the rest of the paper. In my opinion some additional (synthetic) experiments as the one in Fig. 1 can help, as well as classic semi-supervised learning problems with real data.

4. Minor:
	- In the sentence after Eq. 6 there is a typo in the integral.
	- In Eq. 9 should it be $P^*$ or $D^*$?


I think that the paper has a lot of merits and the technical content seems solid. Of course, this is theoretical contribution, but I since it focuses on a particular problem that has potential benefits in applications, I would like to see some additional (synthetic/real-world) experiments.

---

> ### Author Response · Authors · 2022-11-19
> **Reply**
>
>
> We thank the reviewer for taking the time to provide an insightful and constructive review. In what follows, we will reply to the reviewer's questions in detail.
>
> **Lipschitz on Manifolds:** The reviewer asks if a bounded gradient (over the manifold) implies that the function is Lipschitz in the manifold case. The answer is yes, if the function has bounded gradients for, then this bound is the Lipschitz constant. Notice that in the case of manifolds, the norm of the gradient should be bounded for $x\in \mathcal{M}$.
>
> **Point Cloud Laplacian:** This is a very important point. We agree with the reviewers view that a more in-depth explanation of how the point-cloud Laplacian is constructed should be added to the paper. Following the Reviewer’s comment, we have added an ablation study on the construction of the Laplacian; please refer to Appendix I.
>
> We thank the reviewer for this comment.
>
> **Numerical experiments:** This is a good point. Following the reviewers point, we have added an additional experiment with real world data. In section J of the appendix, we considered the problem of predicting the next state of a dynamical system composed of a quadrotor that takes off and flies in circles.
>
> We thank the reviewer for this comment.
>
> **Minor comments (Typo)** We thank the reviewer for finding a typo after equation 6, we fixed it.
>
> **Minor comments (Equation 9)** As a clarification, equation 9 is correct, it should be $P^*$. Note that Proposition 1 proves that the empirical dual problem $\hat D^*$ is an estimator of the primal problem $P^*$.
>
> We would like to thank the Reviewer once again. We hope that our rebuttal will improve their final assessment of our work. We would like to point out that we are more than happy to answer any further questions.

---

### Official Review · Reviewer_HRYo · 2022-10-24

**Confidence:** 3
**Correctness:** 4
**Technical Novelty And Significance:** 3
**Empirical Novelty And Significance:** 2
**Recommendation:** 6

**Clarity, Quality, Novelty And Reproducibility:**

Clarity
--------
I found the paper to be very clear and concise. In particular, the overall structure of paper (going from problem to solution to implementation) is quite nice and makes the paper quite enjoyable to read.

Quality
--------
I found the paper to overall be of pretty high quality. In particular, the contributions are quite nicely developed, and the overall contributions are generally well-formulated.

Novelty
---------
The paper is reasonably novel. In particular, it approaches and established problem and develops a novel methodology to sharpen the constraints.

Reproducibility
-----------------
There was code to reproduce the experiments.

**Strength And Weaknesses:**

Strengths
-----------
* The method is quite nicely developed. In particular, I found the construction to be well-reasoned and nicely created, from the initial optimization problem formulation to the final algorithmic implementation.
* The proposed method seems to be technically correct. In particular, each component is justified nicely.
* The method is quite a novel departure from previous methods, as it handles what feels like a significantly more complex problem formulation. In particular, it merges both lipschitz and manifold constraints.

Weaknesses
---------------
* The experimental section is rather weak/toy.
* The method may not be very applicable. In particular, outside of the standard manifold learning justifications, I question how well this will scale to high dimensional data/large number of data points.

**Summary Of The Paper:**

This paper learns smooth functions (as measured by the Lipschitz constant) on Riemannian manifolds embedded in Euclidean space. Notably, these methods assume the manifold is unknown, presenting a significant technical challenge. The paper then develops several optimization problems and shows to make these tractable through various analyses and perspectives. Empirical results are presented for two tasks.

**Summary Of The Review:**

I lean to accept the paper, mostly due to the technical novelty and very clear exposition. However, I do think that the experimental setup is quite weak, and the applications may be rather limited. However, since this a general trend with manifold learning papers in general, I am willing to somewhat overlook them given the technical novelty.

---

> ### Author Response · Authors · 2022-11-18
> **Reply**
>
>
> We thank the reviewer for the time and effort in providing a useful review. In what follows we will elaborate on the reviewers comments.
>
> **High Dimension/Large number of data points:** This is an excellent point. The reviewer asks how our method scales for high dimensional data, and large numbers of data points. Before providing a detailed explanation, let us clarify that our method does scale.
>
> Notice that our method only requires one scalar per sample (the value of the dual variable $\lambda_i$), and a set of neighbors per sample. The point-cloud Laplacian matrix needs to be calculated only once, and the value of the integral of the point-cloud can be estimated utilizing a batch of samples (i.e. the neighboring data points of each sample, the non-zero elements of the point-cloud Laplacian).
>
> Theoretically, our method scales given that the value of delta in Proposition $3$ depends on the dimension of the manifold $d$, and not the dimension of the data $D$ ($D>d$). This means that as long as there exists a low dimensional manifold, our method does not suffer from the curse of high-dimensions.
>
> We thank the reviewer for this question.
>
> **Experiment section:** This is a good point by the reviewer. The reviewer comments on the lack of real world problems considered in our numerical simulations.
>
> First, let us note that the two-moons dataset, and the navigation controls problem are the only two problems that utilize synthetic data. That is to say, the Ground Robot Error Prediction problem is a real world problem that utilizes real world data. In this experiment, we considered the problem of residual learning in the context of skidding of ground robots, which is a relevant problem, and the data that we utilized is real world data measured from ground robots.
>
> Following the reviewers comment, we have extended our experiment section and we have added another experiment with real world data. We consider the problem of predicting the dynamical system composed of a quadrotor that takes off and flies in circles.  The details of the experiment can be found in section J of the appendix.
>
> We thank the reviewer for this comment.
>
> **Reproducibility:** We would like to mention that the code was provided in the initial submission, and is once again present in the current submission. The code to reproduce all of the results mentioned in the paper can be found in the supplementary material.
>
> We thank the Reviewer once again for their time and valuable comments. We hope that our reply will improve their final assessment of our work. We would be more than happy to take any further questions, or comments.

---

> > ### Comment · Reviewer_HRYo · 2022-11-21
> > **Thank you for the response.**
> >
> > Thank you for your response. I have updated my review accordingly.

---

> > > ### Author Response · Authors · 2022-11-28
> > > **Reply**
> > >
> > > We thank the reviewer for their response. We wanted to bring to their attention that there are no changes to their review. Please let us know if this is a mistake on our side or if there are further comments that we have missed.
> > >
> > > We value your feedback and we do not want to miss any of your comments.

---

### Official Review · Reviewer_FBLh · 2022-10-27

**Confidence:** 2
**Correctness:** 3
**Technical Novelty And Significance:** 2
**Empirical Novelty And Significance:** 2
**Recommendation:** 5

**Clarity, Quality, Novelty And Reproducibility:**

The authors should provide more background and motivation about the problem considered, such that readers could understand it better. The main contributions of this paper should be clarified clearly. The novelty of the paper is good.

**Strength And Weaknesses:**

Strength:

This paper presents a computationally efficient method to learn a Lipschitz continuous function on a manifold, which is a combination of techniques from semi-infinite constrained learning and manifold regularization. Theoretical analysis is also provided.

Weakness:

1. The authors should provide more background and motivation about the problem formulation, i.e., Problem (3). Otherwise, it is difficult for readers to follow.

2. The authors should provide some insights for Proposition 3 about why the integral of the manifold gradient term can be approximated by the point cloud Laplacian term.

3. It's confusing about which problem do the updates (17)-(19) aim to solve.





**Summary Of The Paper:**

This paper studied the problem of learning a Lipschitz continuous function on a manifold. The authors used the Lagrangian dual problem and showed that its empirical version can be an approximation of the primal with statistical consistency. The authors further showed that, by using a weighted point cloud Laplacian, the evaluation of manifold Lipschitz constants can be recast in a more amenable form. The numerical results show the effectiveness of the proposed method.




**Summary Of The Review:**

It is not easy for the reviewer to follow this paper, because of the insufficient background and motivation for the problem formulation considered. The authors should make clearer the main contributions of this paper compared to the previous work.

---

> ### Author Response · Authors · 2022-11-18
> **Reply**
>
>
> We thank the reviewer for their time and effort reviewing our paper. In what follows, we will reply to the comments made by the reviewer.
>
> **Motivation about problem formulation**: The reviewer mentions that more background and motivation about the problem formulation (3) should be provided. Problem (3) is important, because smoothness translates into stability and safety in the case of control, and in the case of machine learning, it improves generalization. Problem (3) aims to find the smoothest function $f_\theta$ that attains a performance of at least $\epsilon$. By smoothest, we mean that the Lipschitz constant or equivalently the norm of gradient of $f_\theta$ over points $x\in\mathcal{M}$ is the smallest. Note that in Problem (3) we aim to find the function whose Lipschitz constant over the manifold $\mathcal{M}$ is the smallest, where manifold $\mathcal{M}$ is defined by the support of the distribution of the data $x\sim p(x)$, and it has a smaller dimension than the input data. That is to say, we consider a $d$ dimensional manifold $\mathcal{M}$, that is embedded in $R^D$, and the ambient dimension is $x\in R^D$.
>
> It should be noted that learning smooth functions was highlighted as a positive strength by the other reviewers. Besides, the clarity of the paper was also a positive remark by the other two reviewers. We thank the reviewer for their opinion.
>
> **Elaboration on Proposition 3**: This is an excellent point. The reviewer mentions that some insights should be provided about Proposition 3, and specifically about why the integral of the manifold gradient can be approximated by a point cloud Laplacian term. It is true that a better elaboration on this should be provided, and we thank the reviewer for pointing it out. We have added the following paragraph to the paper:
>
> > The proof of Proposition 3 relies on two steps. First, we relate the integral of the norm of the gradient over the manifold with the integral of the continuous Laplace-Beltrami operator by virtue of Green's identity. The second part approximates the value of the Laplace-Beltrami operator by the point-cloud Laplacian.
>
> We thank the reviewer for this comment.
>
> **Steps (17)-(19)**: The reviewer states that it is confusing what steps (17)-(19) aim to solve. The short answer is that these steps aim to solve the empirical dual problem given in equation (7).
>
> Note that there are two ways of computing equation (17), one that involves the norm of the gradient, and another one that involves the Laplacian. For steps (18), and (19) there is only one way to compute them. A more in depth explanation of the steps (17)-(19) can be found in Appendix F.
>
> We thank the reviewer for this question.

---

> > ### Comment · Reviewer_FBLh · 2022-12-01
> > **Thank you for your response**
> >
> > I appreciate the response from the authors, which addresses part of my concerns. I would also like to recommend that the authors include more analysis to the experimental section to better show the significance of the results achieved by the new approach.

---

> > > ### Author Response · Authors · 2022-12-06
> > > **Reply**
> > >
> > > We thank the reviewer for their comment.
> > >
> > > The reviewer states that the authors addressed ‘part of’ the reviewer's concerns. The authors are curious to know what part of the original concerns were not addressed by the authors in our previous reply.
> > >
> > > On a separate note, the reviewer suggests that ‘the authors include more analysis to the experimental section’. Let us note that between the original submission, and the last submission, we have added (i) an ablation study on the construction of the point-cloud Laplacian (appendix I), and (ii) a new experiment with real world data (appendix J).
> > >
> > > We would be happy to answer any further questions. We also hope that our response helps improve the Reviewer's assessment of our work.

---

### Decision · Program_Chairs · 2023-01-20

**Decision:**

Reject

**Justification For Why Not Higher Score:**

Although this paper gives an interesting method, the numerical experiments are quite weak. If we consider this is a theoretical paper, the theoretical analysis can be improved mainly by improving the dependency on $\hat{\zeta}$ defined in Assumption 4. Hence, this paper is a little bit under the bar of acceptance.

**Justification For Why Not Lower Score:**

N/A

**Metareview: Summary, Strengths And Weaknesses:**

This paper proposes a method to learn a a Lipschitz continuous function on a manifold that utilizes a weighted Laplacian penalty. The proposed method is theoretically justified by giving a bound on primal-dual gap and evaluating the weighted Laplacian penalty empirically computed. The effectiveness of the proposed method is investigated through some numerical experiment.

The proposed method seems convincing. It is nice that the authors presents a solid theoretical justification on the proposed method. The writing of the paper is clear.
On the other hand, the reviewers raised some concern on the paper. Indeed, some reviewers (HRYo and 2RoL) pointed out the numerical experiments are weak while they appreciate its theoretical value. Some more extensive experiments with (synthetic/real world) data, different tasks and comparison with more baseline methods would strengthen the paper.
Most of the theoretical concerns (mainly its clarity) raised by the reviewers were addressed during the rebuttal period. However, I (AC) personally have some additional concerns on the theoretical part. The proofs of Propositions 1 and 2 seem okay, but Proposition 3 requires some revision. Indeed, the authors try to show the bound by evaluating the probability of the event $C_N$ (Eq.(167) in the supplementary). However, there is no explanation about the choice of $z$ and $\theta$ in the definition of $C_N$. The bound should hold uniformly over the choice of $z$ and $\theta$, but there is no description on this point. Actually, the Hoeffding’s inequality in Eq.173 is provided only for a fixed $\theta$ (it should be noticed that the bound should hold uniformly over $\theta$ instead of arbitrary fixed $\theta$). The same argument applied to Eq.(175) with respect to $z$. To obtain a uniform bound, a standard approach requires to utilize techniques such as Rademacher complexity and covering number, but this paper overlooks this point.

Since this is a rather theoretical paper, the theoretical part is expected to be more solid. Otherwise, the numerical experiments can be much improved. For these reasons, this paper requires revision so that it is accepted in ICLR.

==
Here is another minor concern by AC. The duality gap requires Assumption 4, but the bound in Assumption 4 is given for *all* $\theta$, that means that, the bound should hold uniformly on the model *without* the Lipschitz regularization. Due to this point, the theoretical bound can fail to show the benefit of the Lipschitz regularization because $\hat{\zeta}$ (which should be valid on the entire model) becomes a bottleneck. This can be improved by localizing the bound on the subset of the model with a Lipschitz continuity constraint.